# The Giardial Arginine Deiminase Participates in *Giardia*-Host Immunomodulation in a Structure-Dependent Fashion via Toll-like Receptors

**DOI:** 10.3390/ijms231911552

**Published:** 2022-09-30

**Authors:** Cynthia Fernández-Lainez, Ignacio de la Mora-de la Mora, Sergio Enríquez-Flores, Itzhel García-Torres, Luis A. Flores-López, Pedro Gutiérrez-Castrellón, Paul de Vos, Gabriel López-Velázquez

**Affiliations:** 1Laboratorio de Errores Innatos del Metabolismo y Tamiz, Instituto Nacional de Pediatria, Ciudad de México 04530, Mexico; 2Immunoendocrinology, Division of Medical Biology, Department of Pathology and Medical Biology, University of Groningen and University Medical Center Groningen, 9700 Groningen, The Netherlands; 3Posgrado en Ciencias Biológicas, Universidad Nacional Autónoma de México, Ciudad de México 04510, Mexico; 4Laboratorio de Biomoleculas y Salud Infantil, Instituto Nacional de Pediatria, Ciudad de México 04530, Mexico; 5CONACYT-Instituto Nacional de Pediatria, Secretaria de Salud, Ciudad de México 04530, Mexico; 6Hospital General Dr. Manuel Gea González, Ciudad de México 14080, Mexico

**Keywords:** giardiasis, immune response, inflammation, protein–protein interactions, 3D structure

## Abstract

Beyond the problem in public health that protist-generated diseases represent, understanding the variety of mechanisms used by these parasites to interact with the human immune system is of biological and medical relevance. *Giardia lamblia* is an early divergent eukaryotic microorganism showing remarkable pathogenic strategies for evading the immune system of vertebrates. Among various multifunctional proteins in *Giardia*, arginine deiminase is considered an enzyme that plays multiple regulatory roles during the life cycle of this parasite. One of its most important roles is the crosstalk between the parasite and host. Such a molecular “chat” is mediated in human cells by membrane receptors called Toll-like receptors (TLRs). Here, we studied the importance of the 3D structure of giardial arginine deiminase (GlADI) to immunomodulate the human immune response through TLRs. We demonstrated the direct effect of GlADI on human TLR signaling. We predicted its mode of interaction with TLRs two and four by using the AlphaFold-predicted structure of GlADI and molecular docking. Furthermore, we showed that the immunomodulatory capacity of this virulent factor of *Giardia* depends on the maintenance of its 3D structure. Finally, we also showed the influence of this enzyme to exert specific responses on infant-like dendritic cells.

## 1. Introduction

Human giardiasis is caused by the flagellated protist *Giardia lamblia* and is responsible for diarrheal disease and chronic postinfectious illnesses such as irritable bowel syndrome [1]. Most infections are asymptomatic or mildly symptomatic, such as cramps and mild chronic diarrhea, even though severe complications associated with intestinal malabsorption are also frequent. Resultant diarrhea is thought to be due to nutrient malabsorption, epithelial barrier defects, and ion secretion [2]. However, the mechanisms by which *G. lamblia* causes disease and different symptoms severity are poorly understood.

This parasite has developed many mechanisms to escape from the immune system for growing in the intestine. Although giardiasis is commonly asymptomatic, symptoms such as diarrhea, abdominal pain, nausea, intestinal malabsorption, and weight loss can occur along a broad spectrum. Especially in children, *Giardia* may cause issues as they show a high prevalence of giardiasis. In some cases, it may lead to malnutrition, failure to thrive, stunting, and cognitive impairment [3,4]. Nutrient depletion of the host, caused by the feeding of *Giardia* and secreted cargo by the parasite, might be reasons for these symptoms [5]. 

Alongside other giardial proteins, the arginine deiminase from *G. lamblia* (GlADI) can be found extracellularly in the absence of a signal sequence by a mechanism termed unconventional protein secretion (UPS) [6]. Moonlighting functions (multiple functions for a protein) are often present in UPS cargoes, suggesting that these characteristics are directly linked to each other, as posttranslational modifications seem to influence the function and transport of a protein [5]. GlADI belongs to the arginine dihydrolase pathway for ATP generation in *Giardia* [7,8], which is highly efficient due to the giardial arginine transport system that is 10- to 20-fold higher maximal transport capacity than that of the intestinal epithelial cells (IECs) [9]. 

Along with such a canonical function, GlADI also catalyzes the deamination of the arginine side chain in the conserved CRGKA cytoplasmic tails of the variant-specific surface proteins (VSPs) of *Giardia* [10]. Such VSPs citrullination is proposed to influence antigenic switching and antibody-mediated cell death. Furthermore, the release of GlADI to the medium [11] confers to the parasite the ability to impair the availability of free L-Arg to be used for the production of nitric oxide (NO) by IECs [12]. Since NO inhibits replication of *Giardia* and other microbes, depletion of L-Arg is a recognized strategy used to evade immune effector mechanisms [12,13,14,15].

Among various adaptations *Giardia* uses for coping with the intestinal environment and immune defense, it is valuable in drug design to know how GlADI participates in subverting the intestinal milieu to benefit itself. Regarding this, we previously demonstrated the potential of GlADI as a target for giardiasis treatment by drug repurposing strategies based on cysteine (Cys)-modification mechanisms [16]. It was reported that GlADI is among 16 immunodominant proteins in *Giardia* [17]; however, its role as a virulent factor modulating the immune system into the intestine has not been elucidated enough yet.

In this regard, Banik and colleagues reported that the immunomodulatory effect of recombinant GlADI on human monocyte-derived dendritic cells is attributable to arginine depletion and NH_4_^+^ formation instead of preventing NO formation [18]. Furthermore, Muñoz-Cruz and colleagues reported that citrulline but not ammonium induced activation of rat mast cells [19]. Additionally, they reported that recombinant GlADI still stimulated mast cells in an arginine-free medium, although to a lower extent than in the presence of arginine, indicating that GlADI itself can stimulate mast cells [19]. This latter underlines the importance for GlADI to maintain its 3D structure in the intestine milieu to reach immune cells and to establish a lasting communication between the parasite and the host. 

An essential family of receptors that might be involved in immunomodulation by GlADI is Toll-like receptors (TLRs). TLRs are present in most immune cells involved in responses against *Giardia.* They are known to be modulated by many pathogenic molecules, specific food components, and pharmaceuticals [20,21,22]. The involvement of TLRs in immune responses against *Giardia* is not well studied. Still, it might be important as it might lead to new therapeutical approaches to modulate immune responses against *Giardia* in susceptible groups such as children.

Here, we studied the capacity of GlADI to induce activation of the nuclear factor κB (NF-κB) and activator protein 1 (AP-1) via TLRs in cell lines that endogenously express all human TLRs. Furthermore, the possible impact on TLR-dependent immunomodulation of rabeprazole as a drug against *Giardia* was studied. Spectroscopic fluorescence studies in denaturing conditions analyzed the importance of GlADI 3D structure. To understand the mechanisms by which GlADI exerts immunomodulation through TLRs, we used AlphaFold to create a predicted model of GlADI. We applied it for in silico docking studies to propose its specific binding sites to TLRs. Finally, the impact of GlADI on the cytokine release from immature, infant-like dendritic cells (DCs) was studied.

## 2. Results

### 2.1. GlADI Exerts Immunomodulation by Activating TLRs 

In search of the effects exerted by the direct interaction of GlADI with TLRs, we performed assays to measure the production of NF-κB/AP-1 in THP1-MD2-CD14, a monocyte reporter cell line carrying all TLRs coupled to a SEAP reporter gene. Concentrations of GlADI ranging from 0.05–0.5 μg/mL tended to increase activation of TLRs in a concentration-dependent fashion. On the other hand, GlADI concentrations ranging from 0.5-2 μg/mL had a constant and significantly increased activation of TLRs (*p* < 0.05; Figure 1a). 

We assayed two rabeprazole concentrations to study their effects on the process of TLRs activation exerted by GlADI. The addition of 50 μM rabeprazole did not modify the GlADI-induced TLRs activation. In contrast, a concentration of 750 μM inhibited their activation (Figure 1b). Regarding these experiments, it is remarkable that rabeprazole itself does not activate TLRs with the assayed concentrations (Figure 1b). It is remarkable that only the highest concentration of rabeprazole inhibited the TLRs activation by GlADI. 

### 2.2. GlADI Interacts with the Host Immune System through TLR2 and TLR4 

To determine which specific TLR was activated in THP1 MD2-CD14 cells, the reporter cell lines expressing either human TLRs two, three, four, five, seven, eight, or nine were used. We found that GlADI activated TLR2 in a dose-dependent manner ranging from 1 μg/mL (*p* < 0.001) to 2 μg/mL (*p* < 0.0001) (Figure 2a). Again, rabeprazole had no effect on this TLR but abolished the activation previously exerted by GlADI (Figure 2b).

GlADI with all assayed concentrations also activated TLR4 in a dose-dependent manner from the lowest (*p* < 0.05) to the highest (*p* < 0.0001) (Figure 2c). Interestingly, the activation of TLR4 was significantly more sensitive to GlADI than that of TLR2 (Figure 2a vs. Figure 2c). Furthermore, rabeprazole at a high concentration abolished the activation of GlADI on TLR4. In contrast, the low concentration did not affect this activation process (Figure 2d). GlADI activated neither TLRs three, five, seven, eight, nor nine at any assayed concentrations (Appendix A).

### 2.3. The Forces Maintaining the 3D Structure of GlADI Could Be a Pivotal Factor in Activating TLRs

The protein thermal shift assay showed that GlADI exhibited a stable melting point (~56 °C) in a wide range of pH (pH 5.0 to 9.5) and under a high ionic strength (Table 1). 

To explore whether the tertiary structure of GlADI is involved in the TLRs activation described above, we analyzed the intrinsic fluorescence of this protein by spectroscopic techniques. The quantum yield of GlADI barely varies and is strongly conserved even under extreme denaturing conditions, as can be observed when using increasing concentrations of the chaotropic agent guanidium chloride (GdnHCl) (Figure 3a). The fluorescence emission spectra of GlADI under denaturing conditions showed a maximum of around 340 nm at pH 8.0 (Figure 3a). Such results support that GlADI is highly resistant to denaturing conditions and that the interactions between its constituent amino acids strongly stabilize its 3D structure. 

To correlate the above results showing that the low concentration of rabeprazole did not interfere with the TLRs activation by GlADI, whereas high concentrations did interfere, we assayed the influence of different concentrations of rabeprazole combined with increasing concentrations of GndHCl to determine whether rabeprazole can potentiate denaturing of GlADI. The emission spectra showed that rabeprazole concentrations between 50 and 100 µM could not potentiate GndHCl to denature GlADI (Figure 3b,c). On the other hand, higher concentrations of rabeprazole (500 and 750 µM) strongly potentiated denaturing of GlADI with GndHCl (Figure 3d,e). Furthermore, the data from fluorescence spectra demonstrate the ability of rabeprazole itself to denature GlADI at high concentrations even in the absence of GndHCl (Figure 3f).

Since disulfide bonds can stabilize the 3D structure in proteins, we studied the role they could play in the immunomodulation observed for GlADI. Based on the primary structure of GlADI, the DiANNA 1.1 software (Chestnut Hill, MA, USA) [23] predicted the probability of GlADI forming eight intra-disulfide bonds. Experimentally, we determined only three intra- and possibly one inter-disulfide bond on recombinant GlADI [16]. Furthermore, in the AlphaFold-predicted structure of GlADI, we found 6 Cys residues (C216, C223, C469, C576, C283, C365) per subunit, showing distances propitious to establish three disulfide bonds per subunit (Figure 4). 

As observed in Figure 3a, the 3D structure of GlADI is hardly denatured; therefore, we analyzed its intrinsic fluorescence in the presence of increasing DTT (Dithiothreitol-reducing agent used to break disulfide bonds). In the presence of DTT, the forces that maintain the 3D structure of GlADI (it is, disulfide bonds) were broken, and, finally, the protein was denatured with the addition of 6 M GndHCl (Figure 5a). Furthermore, lower concentrations of GndHCl combined with DTT were able to promote the denaturation of GlADI.

Altogether, our results show the reinforcement of GlADI 3D structure by intra-disulfide bonds. Moreover, on this stable structure, GlADI can activate TLRs. Along with this, both low and high concentrations of rabeprazole can inhibit the enzyme activity of GlADI. Still, only the high ones can break disulfide bonds and impair its 3D structure, with the concomitant inability to activate TLRs. Figure 5b interprets and summarizes the data from the effects of rabeprazole concentrations, protein 3D structure, and TLRs activation. 

### 2.4. Predicted Interactions between the AlphaFold-Predicted Structure of GlADI and TLR2-TLR1

To propose an explanation of the molecular mechanisms that drive the activation effects exerted by GlADI on TLRs, protein–protein docking analyses were performed with the HDOCK server [24]. To that end, the AlphaFold-predicted structure of GlADI and the experimentally determined crystallographic coordinates of human TLR2-TLR1 heterodimer and TLR4-MD-2 heterotetramer (PDB codes: 2Z7X and 3FXI, respectively) were used. 

Molecular docking analysis of TLR2-TLR1 with the dimer of the AlphaFold-predicted structure of GlADI located it in different sites of these receptors. The best-ranked pose of GlADI with human heterodimer TLR2-TLR1 (Figure 6a) had a docking score of −219.48 kcal/mol. The binding affinity (∆G) prediction of the protein–protein complex was −11.8 kcal mol^−1,^ while the dissociation constant (K_D_) predicted was 2.3 × 10^−9^ M at 25 °C. According to this approach, monomer A of GlADI established an interface of interaction with the central and C-term ectodomains of human TLR2 (Figure 6a); of which, an area of 557 Å^2^ comprising six amino acid residues of the monomer A of GlADI interacts with an area of 469 Å^2^ containing 11 amino acid residues of TLR2. Such interactions include one salt bridge, three hydrogen bonds, and 73 non-bonded contacts (Figure 6b). Furthermore, monomer A of GlADI established interaction with TLR1, of which an area of 73 Å^2^ at the region of Arg 153 interacts with an area of 68 Å^2^ comprising three amino acid residues of TLR1. Such interaction formed through 8 non-bonded contacts (Figure 6c) includes important TLR1 residues (V311, F312, and G313), previously reported as part of the ligand binding site [25].

An area of 673 Å^2^ of the monomer B of GlADI comprising 13 amino acid residues interacts with an area of 698 Å^2^ of TLR1, containing 13 amino acid residues. These interactions include one salt bridge, four hydrogen bonds, and 77 non-bonded contacts (Figure 6d). Interestingly, these interactions (Figure 6d, cyan-colored residues) are surrounded by the ligand binding pocket (Figure 6d, magenta-colored residues). 

We analyzed the interacting interfaces between TLR2 and the AlphaFold-predicted GlADI structure, identifying amino acids involved in protein–protein interactions, which are in the region involved in the TLR2 activation process. The GlADI-predicted structure was found to interact with TLR2 mainly through amino acid residues next to the ligand binding site and amino acid residues important for dimerization (Figure 6b, magenta-colored residues). Remarkably, monomer A of GlADI established hydrogen bonds with Y332, S333, and E336 of TLR2 (Figure 6b, blue-colored residues), which are embedded in the ligand binding region where important residues such as L334, V338, I341, and V343 (Figure 6b, magenta-colored residues), are located. These residues were previously reported to interact with acylated lipopeptides [25]. 

### 2.5. Predicted Interactions between the AlphaFold-Predicted Structure of GlADI and TLR4-MD-2

The heterodimer of TLR4 and myeloid differentiation factor 2 (MD-2) recognizes Gram-negative bacteria’s lipopolysaccharide (LPS). Since HEK-Blue-TLR4-MD2-CD14 cells express the TLR4-MD-2 heterodimer, we performed the molecular docking analysis on human TLR4-MD-2 tetramer with the dimer of GlADI (Figure 7). GlADI established five regions of interaction with the TLR4-MD2 tetramer (Figure 7 and Figure 8). The best pose of this interaction had a docking score of −261.63 kcal/mol. The binding affinity (∆G) prediction of the protein–protein complex was −14.6 kcal/mol^−1,^ while the dissociation constant (K_D_) predicted was 1.8 × 10^−11^ M at 25 °C.

An area of 476 Å^2^ comprising six amino acid residues of the monomer A of GlADI interacts with an area of 454 Å^2^ containing seven amino acid residues of TLR4. This interaction region includes two hydrogen bonds, one salt bridge, and 49 non-bonded contacts (Figure 8a). Moreover, monomer A of GlADI established interaction with MD2. Six amino acid residues of GlADI monomer A, which comprised an area of 256 Å^2^, interact with three amino acid residues of MD2, which comprised an area of 289 Å^2^. These interactions include one hydrogen bond and 23 non-bonded contacts (Figure 8b). This interface consists of an important interaction with the K122 residue of MD2, which is part of the ligand binding region. This monomer of GlADI also established interaction with TLR4* by an area of 423 Å^2^ comprising six amino acid residues of the monomer A of GlADI, interacting with an area of 401 Å^2^, containing eight amino acid residues of TLR4* (Figure 8c). The K388 residue of TLR4* is involved in these interactions and is recognized as part of the ligand binding region.

The monomer B of GlADI established interaction with two residues of TLR4 through 2 amino acid residues, contributing to areas of 95 and 96 Å^2^, respectively. This interaction includes four non-bonded contacts (Figure 8d). Furthermore, monomer B of GlADI interacts with MD2 through an area of 357 Å^2^ comprising seven amino acid residues of the monomer B of GlADI interacting with an area of 324 Å^2^ containing nine amino acid residues of MD2. This interaction region includes two hydrogen bonds and 56 non-bonded contacts (Figure 8e). 

### 2.6. GlADI Induces Cytokine Production of DCs

We investigated whether GlADI can influence cytokine production of DCs from umbilical cord blood. We found that GlADI had no significant effect on the production neither of Macrophage Inflammatory Protein 1A (MIP-1A), Interleukin 1 receptor antagonist (IL-1RA), or Interleukin 6 (IL-6) (Figure 9a–c). On the other hand, GlADI caused a significant decrease on the production of chemokine ligand of monocyte chemoattractant protein-1 (MCP-1)/CC (CCL2) (*p* < 0.0001) (Figure 9d) and IL-10 (*p* < 0.05) (Figure 9e), whereas it caused a significant increase in IL-1β (*p* < 0.0001) (Figure 9f) and Tumor Necrosis Factor (TNFα) (*p* < 0.05) (Figure 9g).

## 3. Discussion

In the present study, we demonstrate, until the best of our knowledge for the first time, the direct effect of GlADI on human TLR signaling and report the prediction of its 3D structure based on the AlphaFold artificial intelligence program, as well as the probable mode of interaction of GlADI with human TLRs two and four by molecular docking. Moreover, we show that this immunomodulatory capacity is GlADI 3D structure dependent. We also show that GlADI can influence human infant-like DCs responses. 

Inflammation through TLRs signaling is a protective response of the host to accelerate the healing process against infectious agents. Th1-type cytokines tend to produce the pro-inflammatory responses responsible for killing parasites, whereas Th2 is associated with anti-inflammatory responses. The Th1-biased inflammatory consequences that TLRs exert can also induce fatal pathological outcomes such as septic shock. Additionally, a pathogen modulated TLR signaling can induce a Th2 response, which promotes the progression of the disease. Thus, pro- and anti-inflammatory immune responses must be effectively balanced to restore the homeostasis of the host during and after a pathogenic infection [26].

Various studies on macromolecules show that different TLRs and their downstream pathways can be activated or inhibited in a structure-dependent way [21,27,28,29,30]. The effects of GlADI on TLRs stimulation observed herein are likely to be regulated by factors such as its 3D structure rather than the presence of its metabolic products (citrulline and NH_4_^+^) or depletion of arginine (its substrate). Our findings do not discard the previously described role of the reaction products of GlADI in the immunomodulatory process [18] but contribute to a better understanding of the variety of mechanisms used by *Giardia* to interact with the human immune system. 

Also, as suggested by others [11,18,19], we reinforced that GlADI is a remarkable virulence and pathogenicity factor of *G. lamblia*. Previous studies have found that GlADI can induce immune responses. Muñoz et al. demonstrated the direct role of GlADI in causing the activation of mast cells [19]. However, the authors did not study further implications that structural factors of GlADI could be playing on the observed phenomenon. 

We show that GlADI stimulates the membrane-bounded TLRs two and four. Moreover, our results show that rabeprazole not only inhibits GlADI enzyme activity, as previously reported [16] but also interferes with its immunomodulatory effects on TLRs, likely by breaking GlADI disulfide bonds and inducing its denaturation. The enzyme activity of GlADI is susceptible to rabeprazole since C424 is part of the catalytic triad. Because of this, concentrations of rabeprazole as low as 50 μM can completely inhibit the enzyme [16], whereas high concentrations of rabeprazole start affecting the rest of the cysteine residues and even break disulfide bridges. 

VSPs and other Cys-rich proteins found in *G. lamblia* are exposed to the milieu. They are thought to protect the parasite under the digestive conditions of the upper small intestine [31]. Cysteines are widely distributed in the proteome of *Giardia* [32], and they potentially can bond their sulfur atoms to form the so-called disulfide bonds. These covalent linkages are formed from nonadjacent Cys residues and stabilize the protein 3D structure. In several microorganisms, protein stability often relies on the formation of disulfide bonds to tackle extra cytoplasmic environments [33,34]. Although GlADI is not classified as a Cys-rich protein, its 16 Cys residues per subunit and its ability to be found extracellularly allowed us to assume that it can possess a strong stabilized 3D structure. Our findings of GlADI intrinsic fluorescence under denaturing and reducing conditions support the hypothesis that the disulfide bridges are reinforcing its 3D structure, which could allow it to resist adverse conditions such as those found in the intestinal milieu. 

To gain insight into how the 3D structure of GlADI influences the TLR’s immunomodulatory effect observed, we performed molecular docking studies to identify the sites of interaction of GlADI with TLRs. To that end, GlADI crystallographic structure was needed; however, since it is still unknown, we constructed it by using the neural network AlphaFold, which is the first computational approach capable of predicting protein structures to near experimental accuracy in most cases [35]. For TLR2-TLR1, we show that the AlphaFold-predicted GlADI homodimer structure docked over a previously described region important for ligand binding and dimerization. This prediction suggests that GlADI possibly establishes contacts with TLR1 amino acid residues such as V311, F312, and G313, which participate in the binding of ligands to induce immune activation and cytokine release [25]. Such interactions have an important role in the interaction of TLR1 with agonistic ligands such as the di and tri-acylated synthetic lipopeptides Pam_2_CSK_4_, Pam_3_CSK_4_, and macrophage-activating lipopeptide-2 (MALP-2) and FSL-1 [25,36]. This finding is in line with others since various proteins have been described as newly discovered TLR2 ligands [37]. Altogether these findings allow us to hypothesize that GlADI might be activating the TLR2-TLR1 heterodimer through mechanisms not defined yet, which could be different from those reported for bacterial lipopeptides. Therefore, further studies are needed to understand the mechanisms underlying the stimulation of TLR2 by GlADI. 

Molecular docking supports the possibility of establishing interactions of both TLR4 and MD2 with GlADI through amino acid residues previously described as part of the ligand binding region. This could explain the capacity of GlADI to activate TLR4 more sensitively than that of TLR2. TLR4 is a membrane-bound innate immune receptor protein of 96 kDa that acts as an innate immune sensor against a broad group of invading pathogens, from viruses to multicellular parasites [26,38,39,40]. GlADI is not the first protein from *G. lamblia* with TLR4 activation capacity; others, such as VSPs, have been found to stimulate host innate immune responses in a TLR4-dependent manner [31]. Moreover, TLR4 can bind other ligands in addition to LPS, such as various small molecules and endogenous or exogenous proteins [26,39,41,42,43,44,45,46,47,48,49,50,51], as demonstrated for other diseases such as lymphatic filariasis [40]. Since our in silico proposal predicts protein–protein interactions, it is important to note that molecular docking studies of TLR4 with other proteins, such as the native spike protein of SARS-CoV-2 virus (PDB ID: 6VYB), demonstrated significant binding to TLR4 [52]. Without leaving aside the current lack of experimental evidence about the accuracy of AlphaFold-predicted structures applied in docking predictions, it is interesting to compare the results obtained by using an experimentally resolved 3D structure (as in the case of SARS-CoV-2 spike glycoprotein) and those obtained by us with the AlphaFold-predicted GlADI structure. Compared with those studies, we propose that GlADI might interact with amino acid residues from the leucine-rich repeats (LRR) 3–5 of TLR4, while spike protein is predicted to interact with LRRs 9–13. Furthermore, while spike protein does not interact with residues involved in the canonical activation pathway of TLR4, our results suggest a probability of GlADI doing it (i.e., K388). Additionally, GlADI could interact with K122 from MD2, which is part of the ligand binding region. 

DCs are key players in the intestine immunity present under the epithelial lining of the gastrointestinal tract. They can sample the luminal content by protruding their dendrites into the lumen to distinguish harmful from harmless antigens [53,54,55]. DCs are important initiators of mucosal immune responses both in adults and children. However, the clinical impact of giardiasis seems to be stronger in the first five years of life [56]. Thus we used DCs from umbilical cord blood since they are the most infant-like DCs than other DC models.

Here, the incubation of human infant-like DCs in the presence of GlADI provoked an enhanced release of the pro-inflammatory cytokines TNFα and IL-1β but decreased the release of MCP-1. This would be explained by the activation of the inflammatory response mediated by NF-κB signaling, which is essential for host defense against pathogens or their virulence factors such as GlADI. Accordingly, with our results, studies on rat mast cells showed enhanced release of TNFα after treatment with GlADI [19]. Moreover, a rodent model of infection with *Giardia* showed increased levels of TNFα in plasma [57]. Another study in mice proposed a protective role of this pro-inflammatory cytokine in giardiasis since animals devoid of TNFα showed that peak *Giardia* load levels were around 10-fold higher compared with control mice [58]. However, in the same study, transepithelial resistance was reduced to the same extent despite a much lighter parasite burden in TNFα-responsive mice. Certainly, TNF-α has a very important role in the early control of giardiasis, as previously demonstrated in *Giardia* infection using mouse models [58]. On the other hand, the atrophy of intestinal villi and dysfunction of the gut epithelial barrier detected in biopsy specimens from chronically infected patients with giardiasis can also be observed when healthy intestinal biopsy specimens are treated with TNFα [59,60]. Hence, the affectation on epithelial integrity during giardiasis might be due to parasite factors such as GlADI and host-defense factors such as TNFα. 

We found that the release of the anti-inflammatory cytokine IL-10 was decreased in those DCs incubated in the presence of GlADI. This is in accordance with a previous report where the IL-10 production was found impaired in LPS-activated human monocyte-derived DCs (moDCs), which were exposed to GlADI [18]. Another study reported that children with symptomatic giardiasis had increased mucosal levels of pro-inflammatory cytokines, including TNFα. Such pro-inflammatory cytokines decreased after antigiardial treatment with the concomitant resolution of symptoms and an increase in IL-10 levels [61]. 

Regarding IL-1β, we found that GlADI increased the release of this pro-inflammatory cytokine by DCs. This increase in IL-1β has been suggested to be important for the development and maturation of IL-17A-producing cells [62], which in turn may be essential for the rapid clearance of *Giardia* by humans [63]. 

Ligand binding to TLRs is the first event in a signaling process that allows defenses to be put in place for clearance of the infection and long-term protection via the molecular memory of the adaptive immunity. Considering the state of the art in cytokines release via TLRs stimulation, our results reinforce the proposal that GlADI could be an important factor in giardiasis that impairs the secretion of anti-inflammatory cytokines while enhancing the pro-inflammatory ones. The strengthened 3D structure of GlADI could help the parasite to long-term modulate the host immune response even after the loss of its enzyme activity. Even though AlphaFold is currently the most robust tool for predicting the 3D structure of proteins, it has not replaced the use of experimentally resolved 3D structures for molecular docking. However, it is important to note that it represents a good strategy to accelerate the design of new ways to treat diseases. Further studies are guaranteed to clarify how GlADI could participate in balancing the pathogenic versus host-protective factors during giardiasis. 

## 4. Materials and Methods

### 4.1. Recombinant GlADI Expression and Endotoxin Removal

The expression and purification of GlADI were performed as previously described [16]. Once recombinant GlADI was purified, endotoxin levels were determined with the commercial Pierce™ LAL Chromogenic Endotoxin Quantitation Kit (Thermo Fisher, Waltham, MA, USA) according to the manufacturer’s instructions. Since endotoxin concentration was above 1 EU/mL, it was removed from pure GlADI fraction by membrane affinity using syringe filters (Acrodisc™ Units with Mustang™ E Membrane, PALL, Port Washington, NY, USA). The endotoxin levels were 1.5 EU/mL before removal and 0.02 EU/mL after this process. As previously reported, these latter endotoxin concentrations do not influence the studied cells [64,65]. Nonetheless, to exclude any influence from endotoxin remnants, we added recombinant GlADI to the cells in the presence of 100 μg/mL of the endotoxin-blocker polymyxin B (Invivogen, Toulouse, France).

### 4.2. Reporter Cell Lines

THP1-XBlue™-MD2-CD14 human monocytes were used as reporter cell-line. This cell line endogenously expresses all human TLRs and has been genetically modified with the SEAP inducible reporter gene under the control of NF-κB and AP-1 promoters. It also has an extra insert for the expression of MD2 and CD14 accessory proteins, enhancing TLR signaling [30,66]. Human embryonic kidney cells (HEK-Blue™) expressing either human TLRs 2, 3, 4, 5, 7, 8, or 9 were used. Furthermore, this cell line has a SEAP reporter gene system. It is important to note that the HEK-Blue™ TLR2 cell line also expresses the TLRs 1 and 6. TLR2 forms active heterodimers with TLR1 and TLR6 [66]. All these cell lines were acquired from Invivogen (Invivogen, Toulouse, France). THP1-XBlue™-MD2-CD14 and HEK-Blue™ cell lines were cultured in RPMI-1640 medium with 2 mM glutamine and DMEM medium (Lonza, Basel, Switzerland), respectively. RPMI-1640 contained normocin 100 μg/mL (Invivogen, Toulouse, France) and DMEM medium penicillin/streptomycin 50 U/mL and 50 μg/mL. (Gibco, Leicestershire, UK). Both media were supplemented with 10% heat-inactivated fetal bovine serum (Sigma, St. Louis, MO, USA), sodium bicarbonate 1.5 g/L (Sigma, St. Louis, MO, USA), and sodium pyruvate 1 mM (Biowest, Nuaillé, France). Selection antibiotics (Invivogen, Toulouse, France) were the same as those previously reported [21]. Cell lines were passaged twice weekly and worked at 80% confluency, according to the manufacturer’s instructions.

### 4.3. GlADI-TLRs Interaction Assays

In order to test TLRs activation, THP1-XBlue™-MD2-CD14 and HEK-Blue cell lines were incubated for 24 h at 37 °C and 5% CO_2_, in 96-well plates, at cell densities previously reported [21] in the presence or absence of 10, 50, 100, 500 1000 and 2000 ng/mL of freshly purified GlADI with 100 μg/mL of polymyxin B (InvivoGen, Toulouse, France). In order to challenge rabeprazole with the highest dose of GlADI to evaluate how efficiently this drug inhibited the capacity of GlADI to activate TLRs, the highest GlADI working concentration was added alone or in combination with 50 or 750 μM of rabeprazole (Sigma, St. Louis, MO, USA). Culture medium and agonists for each TLR reported [21] were included as positive and negative controls, respectively. Afterward, TLR activation was determined by quantitation of SEAP production. To this end, 20 μL of supernatant were incubated with Quantiblue™ reagent (Invivogen, Toulouse, France) for 1 h at 37 °C. Change in absorbance was measured at 655 nm in a Bio-Rad Benchmark Plus microplate spectrophotometer reader (Bio-Rad Laboratories B.V, Veenendaal, The Netherlands). Data were normalized relative to the negative control, which was set to 1. 

HEK TLR2 cell lines express TLR2, TLR1, and TLR6 since signaling of TLR2 activation is dependent on TLR2/TLR6 and TLR1/TLR2 interaction. Activation and dimerization of TLR2/1 and TLR2/6 were confirmed by stimulation with the specific agonists Pam3CysSerLys4 (Pam3CSK4) and lipopeptide (FSL-1), respectively.

### 4.4. Dendritic Cells Culturing and Stimulation with GlADI

In order to investigate whether GlADI can influence cytokine production of DCs, human dendritic cells (DCs) were generated from CD34+ progenitor cells harvested from umbilical cord blood (MatTek Corporation, Ashland, MA, USA). DCs were defrosted and incubated with a maintenance medium containing cytokines (DC-MM, MatTek Corporation, Ashland, MA, USA) according to the manufacturer’s instructions, for 24 h at 37 °C, in 96-well plates at a density of 70 × 10^4^ cells/well to allow them to attach to the well bottom. After 24 h of incubation, DCs were incubated for 48 h at 37 °C and 5% CO_2_ in a culture medium containing 500 ng/mL of freshly produced GlADI. In another set of experiments, to challenge DCs with the minimal dose of GlADI that activated TLRs, DCs were incubated with 500 ng/mL of GlADI in combination with 50 μM of Rabeprazole. For all cases, cell supernatants were collected and stored at −80 °C for their subsequent use. Six independent assays were performed. DCs treated with 10 ng/mL of lipopolysaccharide (LPS) from *E. coli* K12 (Invivogen, Toulouse, France) served as positive controls, and cells incubated only with a culture medium were used as negative controls.

### 4.5. Quantitation of Dendritic Cells Cytokines Production

In order to measure the levels of cytokines TNFα IL-10, IL-6, IL-1β, MCP-1/CCL2, and MIP-1α/CCL3, a magnetic Luminex ^®^ Assay (R&D systems, Biotechne, Minneapolis, MN, USA) was used according to manufacturer’s instructions. Briefly, serial dilutions of each cytokine standard were prepared. A mixture of magnetic microbeads with immobilized antibodies was added to a 96-well plate, followed by the addition of undiluted DC supernatants or standards. The plate was incubated overnight at 4 °C in constant shaking. After three washing steps, detection antibodies were added, and the plate was incubated for 30 min at room temperature under continuous shaking. After three more washing steps, the plate was incubated with streptavidin-PE for 30 min in constant shaking. Finally, after washing, magnetic microbeads were resuspended in 100 μL/well of wash buffer, followed by the plate analysis using a Luminex 200 system (Biotechne, Abingdon, UK). Cytokine data were analyzed with xPONENT 4.2 software (Luminex Corporation, ‘s-Hertogenbosch, The Netherlands). 

### 4.6. Protein Thermal Shift Assay 

The thermal shift assay was performed using a Protein Thermal Shift™ Dye Kit (×1000; Thermo Fisher Scientific; Waltham, MA, USA). Each reacted sample (60 µL) was mixed with SYPRO Orange dye (×2) (60 µL), and 20 µL was dispensed into a tube. Using the StepOnePlus real-time PCR system (Applied Biosystems, Carlsbad, CA, USA), the sample was thermally denatured by increasing the temperature from 25 °C to 99 °C at a rate of 0.16 °C/min; the fluorescence intensity was measured. Data were analyzed using Protein Thermal Shift Software v1.0 (Applied Biosystems; Foster City, CA, USA) to determine the Tm value. The Tm value was used based on the Boltzmann fitting of the fluorescence/temperature raw data (TmB value).

In the thermal shift assay, when a fluorescent dye is added to a protein and heated, the dye binds to exposed hydrophobic sites, which causes it to fluoresce, and the change in fluorescence intensity can be measured to analyze the structural changes in the protein.

### 4.7. Fluorescence Emission Spectra

The intrinsic fluorescence is used to follow the exposure to solvent of aromatic amino acid residues that occurs during the protein unfolding, which is accompanied by a decrease in quantum yield and a redshift of the maximal emission wavelength. Fluorescence experiments were performed using a Perkin–Elmer LS55 spectrofluorometer (Perkin-Elmer, Waltham, MA, USA) at 25 °C and a protein concentration of 65 μg/mL. The intrinsic fluorescence of the enzymes was determined at an excitation wavelength of 295 nm, and the emission spectra were recorded from 310 to 500 nm, with an integration time of 1s, using excitation and emission slits of 3.5 nm each. Each spectrum was the average of three scans with two experimental repetitions. The spectra of blanks were subtracted from those containing protein.

### 4.8. Prediction of the Three-Dimensional (3D) Structure of GlADI

Since the experimentally determined 3D structure of GlADI is still unavailable, we used its primary sequence (UniProt code Q27657) to predict the 3D atomic coordinates of folded protein structure. AlphaFold Monomer v2.0 pipeline [35] was used in the Colab server (https://colab.research.google.com/github/sokrypton/ColabFold/blob/main/AlphaFold2.ipynb (accessed on 2 July 2022)). The predicted protein structure started from their sequence, using a slightly simplified version of AlphaFold v2.0 without selecting a specific existing structural template. Notably, the obtained protein structure of the monomer includes the C-terminal end, which is absent in any other homology models previously described [16,67,68]. This latter is relevant since GlADI is the enzyme of its type that shows the most extended C-terminal domain described until today [68]. The dimer was constructed with PyMOL [69] using crystallographic symmetry based on the atomic coordinates of arginine deiminase from *Pseudomonas auruginosa* (PDB code: 2A9G). 

### 4.9. Prediction of the Probably Binding Mode of GlADI to TLRs by Using the AlphaFold-Predicted Structure and Molecular Docking

Before docking analyses, YASARA Energy Minimization Server [70] was used to attain a minimum energy arrangement of the AlphaFold-predicted 3D structure of GlADI. The HDOCK server for integrated protein–protein docking [24] was used to carry out molecular docking between the dimer of the AlphaFold-predicted structure of GlADI and TLR2-TLR1 (PDB code: 2Z7X) or TLR4-MD2 (PDB code: 3FXI) dimers. In our analyses, we established GlADI as the ligand molecule while TLR2-TLR1 or TLR4-MD2 dimers were established as receptor molecules. Results of HDOCK simulations were confirmed by submitting HDOCK-generated protein–protein complexes to PRODIGY [71].

### 4.10. Statistical Analyses

Data were analyzed with GraphPad Prism™ software (version 8.2.1 for Windows™, San Diego, CA, USA). The normal distribution of data was assessed with the Shapiro–Wilk test. Normally distributed data were analyzed with one-way ANOVA followed by Dunnett’s multiple comparisons adjustment. Non-parametric distributed data were analyzed with the Mann–Whitney U test or Friedman test, followed by Dunn’s multiple comparisons adjustment test. Results are expressed as mean ± SD or the median and interquartile range (IQR) for data with parametric and non-parametric distribution, respectively. A *p*-value < 0.05 was statistically significant (* *p* < 0.05, ** *p* < 0.01, *** *p* < 0.001, **** *p* < 0.0001).

## Figures and Tables

**Figure 1 ijms-23-11552-f001:**
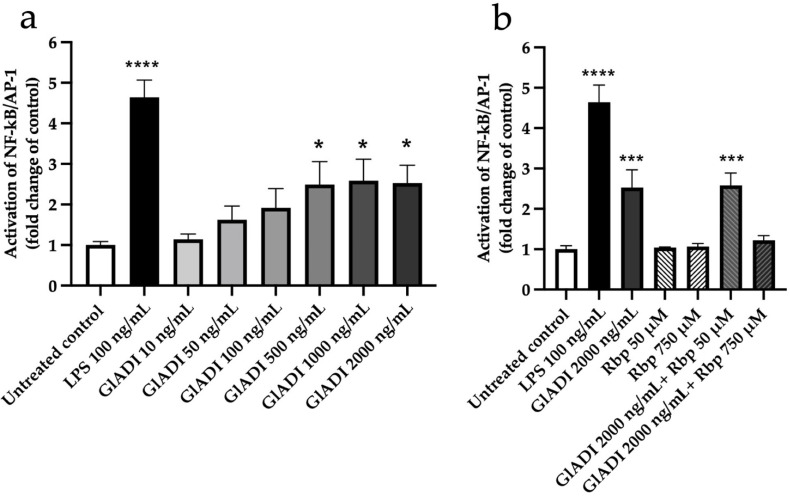
NF-κB/AP-1 activation in THP1-MD2-CD14 reporter cells. (**a**) Cells were stimulated with rising concentrations of GlADI and (**b**) with rabeprazole (Rbp) or the highest concentration of GlADI mixed with Rbp. Activation of NF-κB/AP-1 is presented as the untreated control fold change. Results represent the median with the interquartile range of at least three independent experiments, with three technical replicates. Statistical significance levels compared to the untreated control were determined by the Friedman test (non-parametric statistical test), followed by the Dunn’s multiple comparisons test (post hoc test). A *p*-value *<* 0.05 was considered as statistically significant (* *p <* 0.05, *** *p <* 0.001, **** *p <* 0.0001).

**Figure 2 ijms-23-11552-f002:**
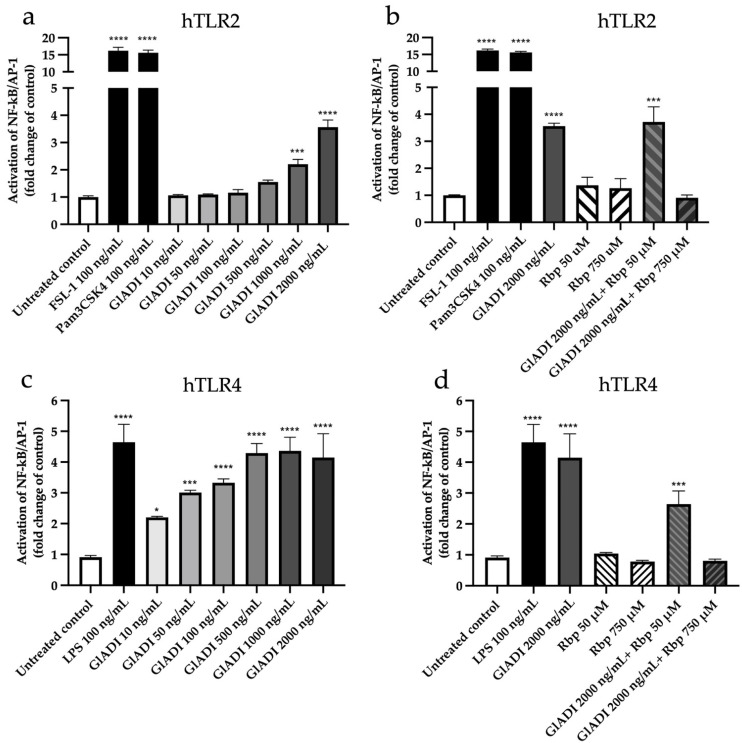
Activation effects of GlADI on HEK-Blue™ reporter cell lines. Each cell line was incubated for 24 h with rising concentrations of GlADI, with rabeprazole (Rbp), or with a mix of GlADI 2000 ng/mL and Rbp. Afterward, NF-κB/AP-1 release was determined. Activation of TLRs is presented as a fold change of the untreated control. NF-κB/AP-1 activation effect of GlADI (**a**,**c**) and the Rbp (**b**,**d**) on human TLR2 (**a**,**b**) and human TLR4 (**c**,**d**), respectively. Appropriate agonists for each TLR served as positive controls. At least five independent assays, each one with three technical replicates. These data were normally distributed. Therefore, the results are represented as the mean ± SD. Statistical significance levels compared to the negative control were determined by one-way ANOVA with Holm-Sidak’s multiple comparisons test. A *p*-value *<* 0.05 was considered as statistically significant (* *p <* 0.05, *** *p <* 0.001, **** *p <* 0.0001).

**Figure 3 ijms-23-11552-f003:**
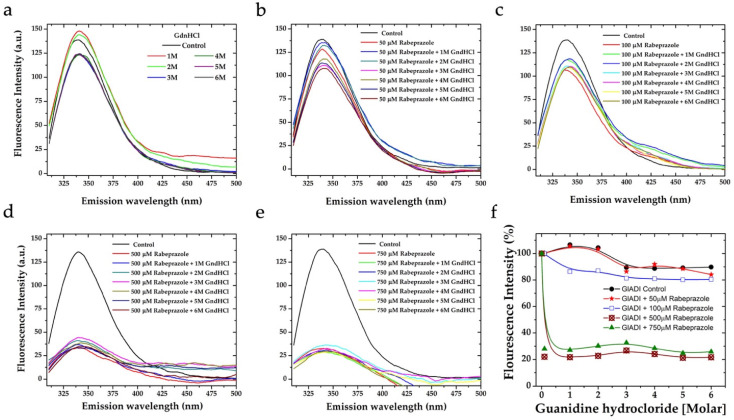
Fluorescence emission spectra of GlADI at rising concentrations of the chaotropic agent guanidine hydrochloride (GdnHCl). The intensity of intrinsic fluorescence of GlADI is almost not decreased by increasing concentrations of GdnHCl tested in a range of denaturing concentrations (**a**), even when rabeprazole 50 μM (**b**) or 100 μM (**c**) are added. The intensity of intrinsic fluorescence of GlADI is strongly decreased when rabeprazole 500 μM (**d**) or 750 μM (**e**) are added. The overall data show the effects of low and high concentrations of rabeprazole on the intrinsic fluorescence of GlADI in the presence or absence of GdnHCl (**f**).

**Figure 4 ijms-23-11552-f004:**
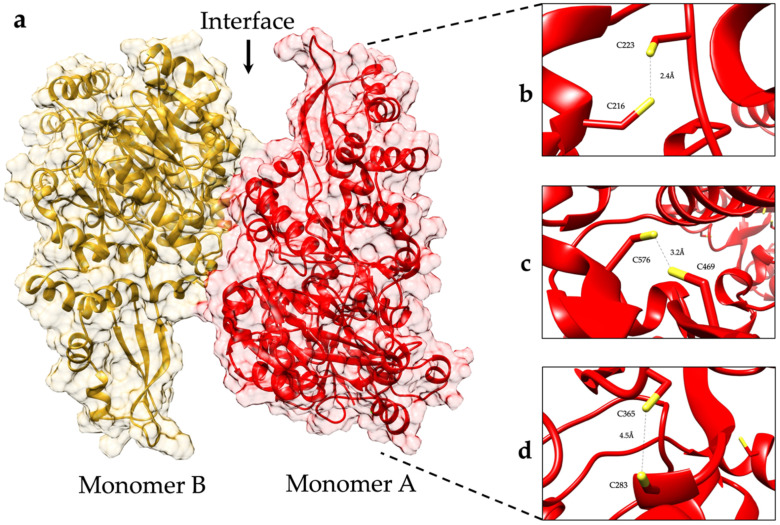
Ribbons and surface representation of the AlphaFold-predicted structure of GlADI. (**a**) The homo dimer with two identical subunits is the described biological active form of GlADI. (**b**–**d**) Based on the distance between their sulfur atoms, three pairs of cysteines might be forming disulfide bonds.

**Figure 5 ijms-23-11552-f005:**
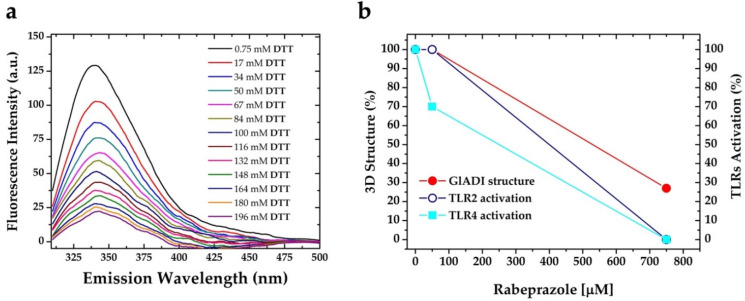
Relationship between 3D structure of GlADI and activation of TLRs. Fluorescence emission spectra of GlADI added with 6 M GndHCl dropped off by increasing concentrations of DTT (disulfide disrupting agent) as a reflection of its denaturing (**a**). The 3D structure stability of GlADI correlates with low concentrations of rabeprazole and activation of TLRs. In contrast, a lack of 3D structure stability correlates with a lack of TLRs activation and high concentrations of rabeprazole (**b**).

**Figure 6 ijms-23-11552-f006:**
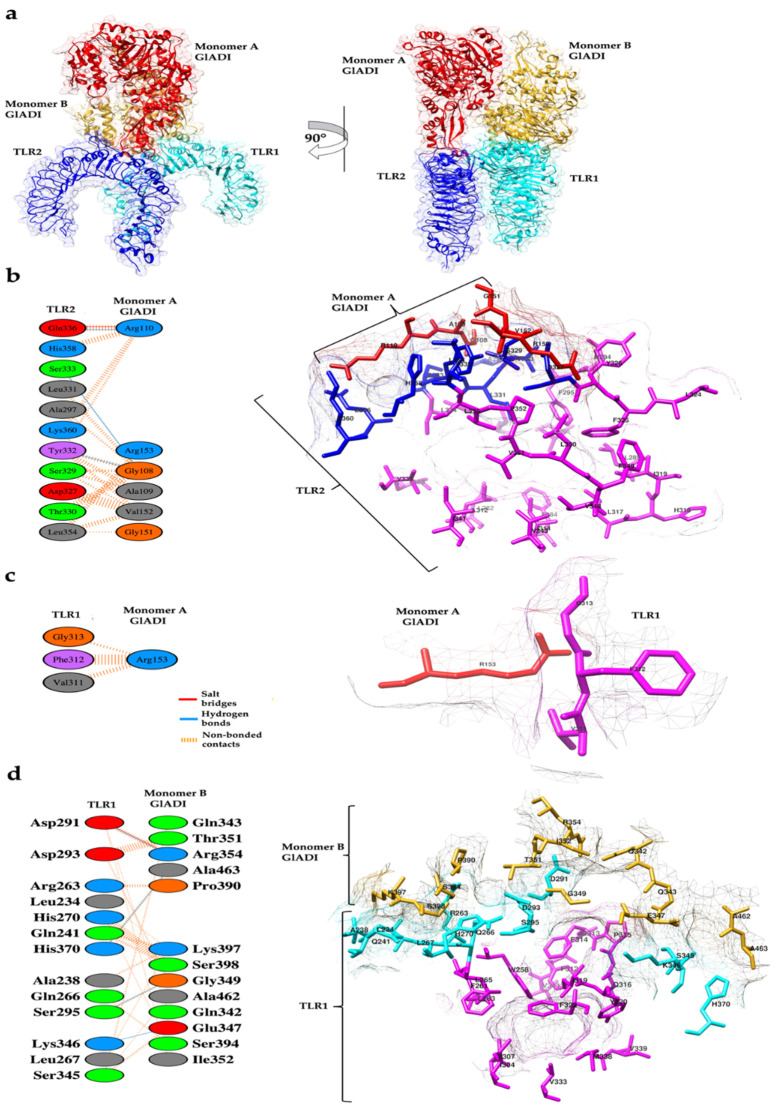
Predicted interactions between AlphaFold-predicted structure of GlADI and human TLR2-TLR1 heterodimer. (**a**) Two views of the predicted protein–protein interactions of GlADI with the ectodomains of human TLR2-TLR1. (**b**) Amino acid residues interacting between GlADI monomer A (red-colored, right side) and TLR2 (blue-colored, right side). (**c**) Amino acid residues interacting between GlADI monomer A (red-colored, right side) and TLR1 (magenta-colored, right side). (**d**) Amino acid residues interacting between GlADI monomer B (gold-colored, right side) and TLR1 (cyan-colored, right side). Amino acid residues magenta-colored (right side) are part of the ligand binding site of the TLR2-TLR1 heterodimer. Interacting amino acid residues are shown (left side). Structures displayed with UCSF Chimera. Docking was performed with HDOCK. Interactions analyses were made with PDBSum.

**Figure 7 ijms-23-11552-f007:**
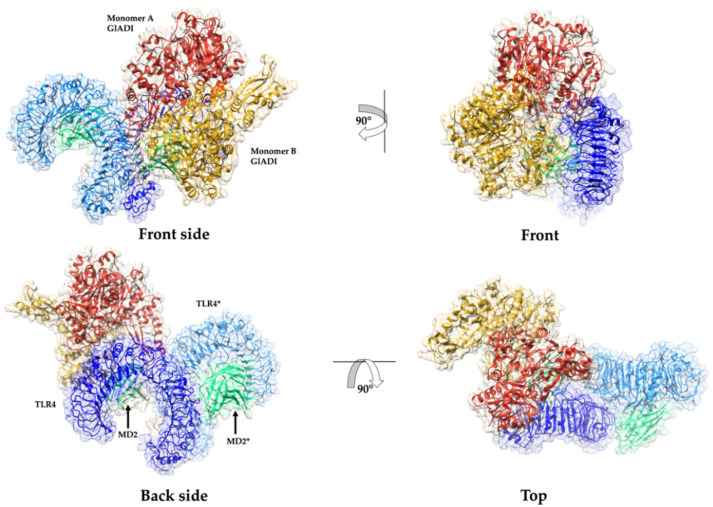
Predicted interactions between AlphaFold-predicted structure of GlADI and the ectodomain of human TLR4 in complex with MD-2. Ribbons combined with surface representation show four different views of the docking prediction between GlADI and TLR4-MD2 tetramer. Structures displayed with UCSF Chimera.

**Figure 8 ijms-23-11552-f008:**
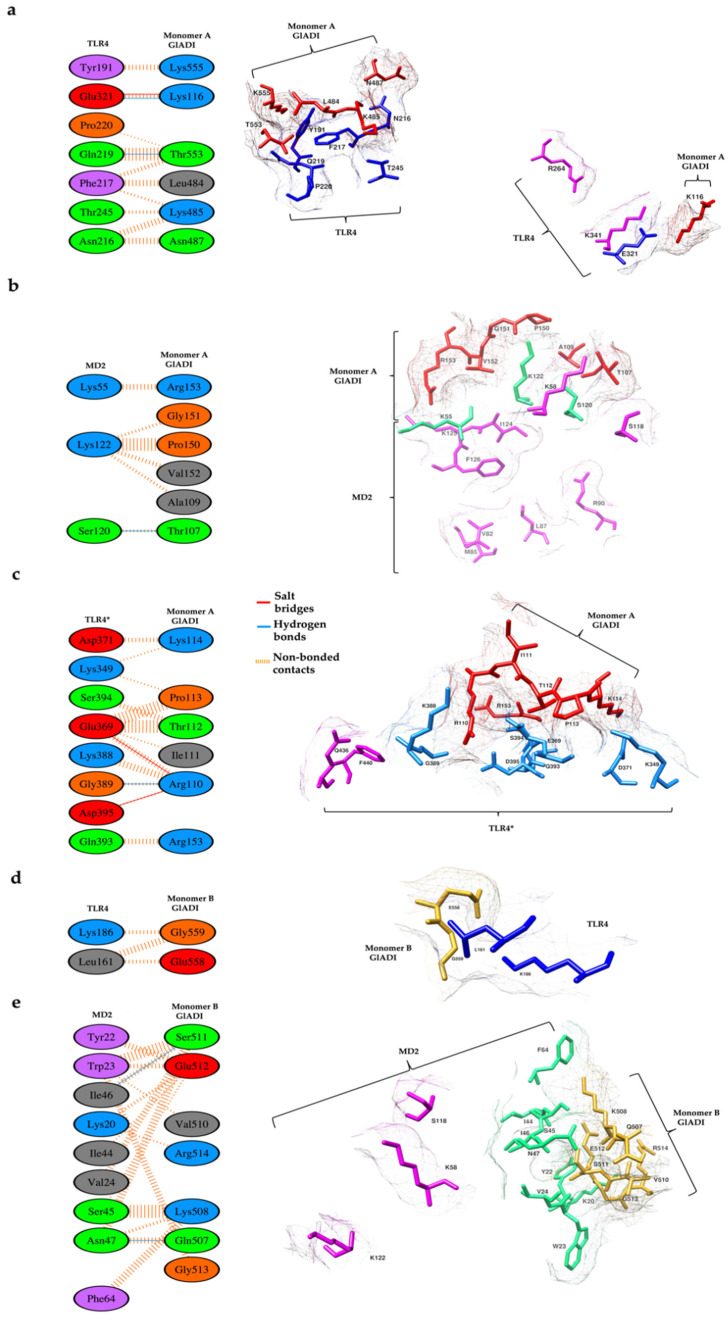
Amino acid residues proposed to interact between AlphaFold-predicted structure of GlADI and the ectodomain of human TLR4 dimer in complex with MD-2. (**a**) Protein–protein interaction of monomer A of GlADI (red-colored, right side) with TLR4 (blue-colored, right side) and the interactions between their amino acid residues (left side). (**b**) The amino acid residues of monomer A of GlADI (red-colored, right side) interact with MD2 (green-colored, right side). (**c**) The amino acid residues of monomer A of GlADI (red-colored, right side) interact with TLR4*(cyan-colored, right side). (**d**) The amino acid residues of monomer B of GlADI (gold-colored, right side) interact with TLR4 (blue-colored, right side). (**e**) The amino acid residues of monomer B of GlADI (gold-colored, right side) interact with MD2 (green-colored, right side). Amino acid residues magenta-colored (right side) are part of the ligand binding site of the TLR4-MD2 tetramer. Interacting amino acid residues are shown (left side). Structures displayed with UCSF Chimera. Docking was performed with HDOCK. Interactions analyses made with PDBSum.

**Figure 9 ijms-23-11552-f009:**
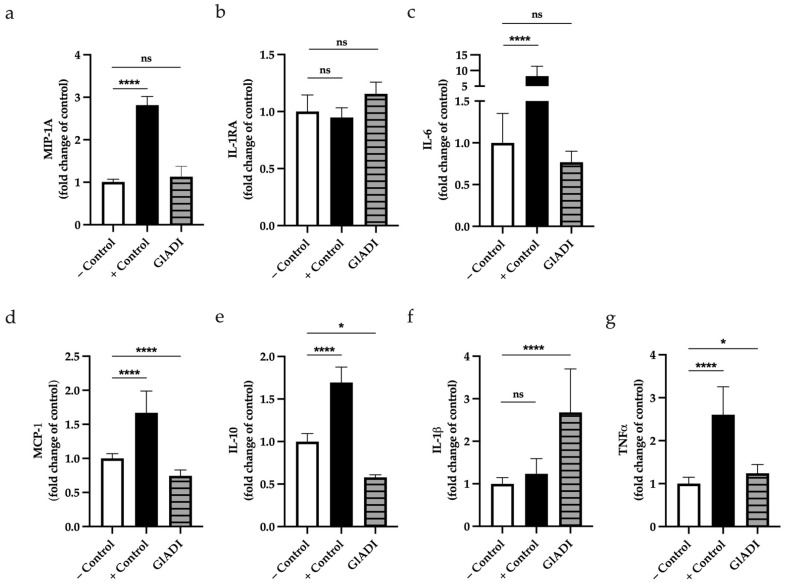
Cytokine production by DCs pre-treated with GlADI 500 ng/mL. GlADI had no effect on the production of (**a**) MIP-1A), (**b**) IL-1RA, and (**c**) (IL-6), whereas it caused a significant decrease in the production of (**d**) MCP-1 and (**e**) IL-10. Significant increases in the production of (**f**) IL-1β and (**g**) TNFα are shown. Results are shown as fold change of the untreated control. Positive and negative control experiments are shown as − Control and + Control, respectively. A *p*-value *<* 0.05 was statistically significant (* *p <* 0.05, **** *p <* 0.0001, ns = no significant).

**Table 1 ijms-23-11552-t001:** Thermal shift assays on GlADI varying pH and salinity.

	Melting Temperatures of GlADI (°C)							
[NaCl] mM	pH 5.0	pH 6.0	pH 6.5	pH 7.0	pH 7.5	pH 8.0	pH 9.0	pH 9.5
0	55.98	59.14	55.88	55.64	60.34	58.96	50.04	56.99
200	49.82	60.94	60.97	63.01	70.56	56.4	59.74	56.24
400	57.04	50.59	47.17	48.12	53.57	52.05	51.93	56.84

## Data Availability

Data are contained within the article.

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
