# Peer review of "The Giardial Arginine Deiminase Participates in Giardia-Host Immunomodulation in a Structure-Dependent Fashion via Toll-like Receptors"

_ijms, 2022, doi:10.3390/ijms231911552_

Round 1
Reviewer 1 Report
The ability to predict protein structures accurately based on the constituent amino acid sequence has a wide variety of benefits in the life sciences space including accelerating advanced drug discovery and enabling better understanding of diseases. In the peer-reviewed manuscript: “The giardial arginine deiminase participates in Giardia-host immunomodulation in a structure-dependent fashion via Toll-Like Receptors”, Authors tried to prove using computional biology technics the hypothesis included in the title. In my opinion, the conclusions drawn should be much more cautious than they are presented in the paper (examples l. 411, 412, 425, 430, 434 and some more). My doubt are bound with using AlphaFold 2 by Authors, however not generally to predict 3D structure of GlADI, but using obtained in such a way data in the molecular docking analysis and on the basis of it drawing far-reaching conclusions. This is a relatively new tool and is mostly dedicated for protein 3D structure prediction. How the amino acid sequence can determine the 3-D structure is already highly challenging, let alone predict the interaction of various proteins which 3D structure had not been determined experimentally (even AlphaFold 2 doesn't have a 100% match of experimentally determined 3D structure).
It was noted by Stephen Curry that while the resolution of AlphaFold 2's structures may be very good, the accuracy with which binding sites are modelled needs to be even higher: typically molecular docking studies require the atomic positions to be accurate within a 0.3 Å margin, but the predicted protein structure only have at best an RMSD of 0.9 Å for all atoms. So AlphaFold 2's structures may only be a limited help in such contexts. Thus, it is not yet clear to what extent structure predictions made by AlphaFold 2 will hold up for proteins bound into complexes with other proteins and other molecules.
It is known that the 3-D structure determines the biological function of the protein. However, many other factors play crucial role in it. Since the model AlphaFold 2 only predicts the main peptide chain, not the structures of missing co-factors, metals, and co- and post-translational modifications it is another reason why the presented in the manuscript conclusions should be more balanced.
In my opinion, the Authors should discuss among others above mentioned limitations of the AlphaFold 2 model. In addition, the Authors should very clearly articulate that this type of research may speed up finding the right path to develop new strategies and treat various diseases, but it will not replace them yet. At one point in the manuscript, the need for further research was advisable, but in my opinion this statement was too laconic. The next step to prove the Authors hypothesis should be experimental determination of 3D structure of GlADI and the repetition of molecular docking analysis with such data.
Other detailed comments:
l. 38-42; I recommend organizing the first paragraph; once it is said that giardiasis is “an inflammation-driven diarrhea” and then “giardiasis is not associated with overt inflammation”. It would be better to briefly describe the clinical forms first, and only then refer to the mechanisms.
l.102-118 In my opinion should or could be part of the Introduction chapter rather than the Results. Moreover in this part the Authors need to correct the style, sentences are too complex and not all information needed, furthermore there is no cause-and-effect relationship for some information
I would propose to start Results with l. 118 “In search of the effects exerted by direct…
l. 121 In the given range (0,5-2 µg/ml), the effect of GlADI is constant, while in the lower 0.05 - 0.5, this effect is dependent of the concentration
l.123-133 Information from previous surveys and their comparison with current surveys should be part of the Discussion, not the results; this would give a clearer picture of the results currently obtained
l. 135 I propose to omit “expressing all TLRs”; this information can be found in Materials and methods chapter
l.136-137 please simplify and remove such detailed information from the signature
l.143-144 – please remove the part from probably Template about Table 1
What was the reason of GlADI 2000 ng/ml dose selection in the experiment with rabeprazole (Fig 1), while already 500 mg/ml gave the same activation. And if the Authors, for some reason, have chosen that dose why they used 500 mg/ml in DCs stimulation for cytokine production.
Fig. 7 should be placed in the main text near to the first time they are cited, so in the chapter 2.5
l.434 - In relation to my initial comments; The authors are not entitled to compare the results of their research with those contained in [52] using the statement "similar to what we found" since 3D structure of the wild‐type SARS‐CoV‐2 spike glycoprotein (PDB ID: 6VYB) was retrieved from the RCSB Protein Data Bank, which was determined de novo by using cryo‐electron microscopic technique with a resolution of 3.20 Å and not AlphaFold 2
l.441-447 This fragment is about hypotheses, speculations and not conclusions from the obtained results
Author Response
Responses are red written.
Reviewer 1
The ability to predict protein structures accurately based on the constituent amino acid sequence has a wide variety of benefits in the life sciences space including accelerating advanced drug discovery and enabling better understanding of diseases. In the peer-reviewed manuscript: “The giardial arginine deiminase participates in Giardia-host immunomodulation in a structure-dependent fashion via Toll-Like Receptors”, Authors tried to prove using computional biology technics the hypothesis included in the title. In my opinion, the conclusions drawn should be much more cautious than they are presented in the paper (examples l. 411, 412, 425, 430, 434 and some more). My doubt are bound with using AlphaFold 2 by Authors, however not generally to predict 3D structure of GlADI, but using obtained in such a way data in the molecular docking analysis and on the basis of it drawing far-reaching conclusions. This relatively new tool is mostly dedicated to protein 3D structure prediction. How the amino acid sequence can determine the 3-D structure is already highly challenging, let alone predict the interaction of various proteins which 3D structure had not been determined experimentally (even AlphaFold 2 doesn't have a 100% match of experimentally determined 3D structure).
Stephen Curry noted that while the resolution of AlphaFold 2's structures may be very good, the accuracy with which binding sites are modelled needs to be even higher: typically molecular docking studies require the atomic positions to be accurate within a 0.3 Å margin, but the predicted protein structure only have at best an RMSD of 0.9 Å for all atoms. So AlphaFold 2's structures may only be a limited help in such contexts. Thus, it is not yet clear to what extent structure predictions made by AlphaFold 2 will hold up for proteins bound into complexes with other proteins and other molecules.
It is known that the 3-D structure determines the protein's biological function. However, many other factors play crucial role in it. Since the model AlphaFold 2 only predicts the main peptide chain, not the structures of missing co-factors, metals, and co- and post-translational modifications it is another reason why the presented in the manuscript conclusions should be more balanced.
In my opinion, the Authors should discuss among others above mentioned limitations of the AlphaFold 2 model. In addition, the Authors should very clearly articulate that this type of research may speed up finding the right path to develop new strategies and treat various diseases, but it will not replace them yet. At one point in the manuscript, the need for further research was advisable, but in my opinion this statement was too laconic. The next step to prove the Authors hypothesis should be experimental determination of 3D structure of GlADI and the repetition of molecular docking analysis with such data.
Based on the comments, we rewrote the concluding sentences more cautiously throughout the manuscript. Also, we mentioned the limitations of the AlphaFold 2 model but emphasized its utility in accelerating the designing of new ways to treat diseases.
Other detailed comments:
l. 38-42; I recommend organizing the first paragraph; once it is said that giardiasis is “an inflammation-driven diarrhea” and then “giardiasis is not associated with overt inflammation”. It would be better to briefly describe the clinical forms first and then refer to the mechanisms.
We agree. The introduction was rephrased as suggested.
l.102-118 In my opinion should or could be part of the Introduction chapter rather than the Results. Moreover in this part the Authors need to correct the style, sentences are too complex and not all information needed, furthermore there is no cause-and-effect relationship for some information.
We agree. The unnecessary information was deleted, and the paragraph was rephrased.
I would propose to start Results with l. 118 “In search of the effects exerted by direct…
We agree. The results section starts with the recommended sentence.
l. 121 In the given range (0,5-2 µg/ml), the effect of GlADI is constant, while in the lower 0.05 - 0.5, this effect is dependent on the concentration.
We agree. A paragraph according to this comment was included (l.109-112).
l.123-133 Information from previous surveys and their comparison with current surveys should be part of the Discussion, not the results; this would give a clearer picture of the results currently obtained.
We agree. Information from previous surveys and their comparison with current surveys were deleted from the results section and incorporated in the discussion section.
l. 135 I propose to omit “expressing all TLRs”; this information can be found in Materials and methods chapter
We agree. The sentence “expressing al TLRs” was deleted.
l.136-137 please simplify and remove such detailed information from the signature
We agree. We simplified and removed detailed information.
l.143-144 – please remove the part from probably Template about Table 1
Done.
What was the reason of GlADI 2000 ng/ml dose selection in the experiment with rabeprazole (Fig 1), while already 500 mg/ml gave the same activation. And if the Authors, for some reason, have chosen that dose why they used 500 mg/ml in DCs stimulation for cytokine production.
We decided to challenge rabeprazole with the highest dose of GlADI to evaluate how efficiently this drug inhibited the capacity of GlADI to activate TLRs. On the other hand, we decided to challenge DCs with a minimal dose of GlADI that activated TLRs since DCs are very reactive to external stimuli. We incorporated these explanations into the corrected manuscript in the methods section (l. 522-524 and l. 546-548).
Fig. 7 should be placed in the main text near to the first time they are cited, so in the chapter 2.5.
We agree. Fig. 7 was relocated as recommended.
l.434 - In relation to my initial comments; The authors are not entitled to compare the results of their research with those contained in [52] using the statement "similar to what we found" since 3D structure of the wild‐type SARS‐CoV‐2 spike glycoprotein (PDB ID: 6VYB) was retrieved from the RCSB Protein Data Bank, which was determined de novo by using cryo‐electron microscopic technique with a resolution of 3.20 Å and not AlphaFold 2
We agree. The paragraph was rewritten based on this comment (l.429-435).
l.441-447 This fragment is about hypotheses, speculations and not conclusions from the obtained results.
We agree, and the speculations were deleted.
We thank the reviewer for his/her comments which substantially improved our manuscript.

Reviewer 2 Report
1. Too much discussion prior to the description of the results. For example: Line 102-114 can be incorporated in the discussion. Same for the other segments of result as well.
2. Dis the authors examine glycosylation in GIADI?
3. Authors are advised to check the comparative effect of pharmacological and SiRNA-mediated inhibition of the TLRs in GIADI-induced immunomodulatory consequences.
4. For DC experiments, positive and negative control experiments should be incorporated.
5. Please follow and cite the following articles in the discussion related to parasite -induced TLR activation: 10.1016/j.bjid.2015.10.011, 10.1093/infdis/jix067, 10.1038/s42003-019-0392-8, 10.1111/pim.12389
Round 2
Reviewer 2 Report
Authors have addressed all the issues.